# "Christ Is Speaking": The Psalms as the Grammar of Augustine's Sermons

**Matthew D. Love** 

Harding School of Theology, Memphis, TN 38117, USA; mlove1@harding.edu

**Abstract:** The Psalms saturated Augustine's sermons. He believed they were God's words to the church as inspired Scripture, and the church's words to God as prayer and praise. In the Psalms, he saw *kenosis*, the downward-directed God in Christ who emptied himself to take on human nature to stand in solidarity with the church and creation. He saw, too, the possibility of deification, the upward-directed church in Christ raised to share in the divine nature. Furthermore, Augustine believed that Christ himself spoke in the Psalms so that in them the church could hear his voice and come to know its own voice. In this essay, I examine why Augustine cherished the Psalms, and I consider how this might inspire contemporary preachers to cherish them and preach them. Moreover, I offer Augustine's Christocentric preaching of the Psalms as a paradigm for how preachers might facilitate Christological formation among their congregants.

**Keywords:** Augustine; Psalms; preaching; prosopological exegesis; prosopopoeia; *kenosis*; deification; *totus Christus*

## 1. Introduction

Augustine scholar Carol Harrison relays a popular saying attributed to Augustine by scholars, that *St. Augustine wrote not in Latin but in the Psalms* (Harrison 2011, p. 208). He preached the same way: not in Latin but in the Psalms. One finds the Psalms throughout his published works. The Psalms functioned as a type of grammar for Augustine through which he wrote and preached, a system and structure that gave form and support to his thought.[1] Even more foundationally, however, the Psalms functioned as a grammar by which Augustine understood himself and the church.

This essay describes Augustine's love of the Psalms, claiming that Augustine loved the Psalms chiefly because he heard Christ's voice in them. Augustine read these texts within a tradition that read them not merely as though they were *about* Christ, but as if they were *spoken by* Christ. In discerning Christ's voice in the Psalms, Augustine learned a language for himself and for all the church. The Psalms are unique in the canon for the ways they comprise the full gamut of life: base words of human agony and anger, and celestial words of promise and praise. These texts give voice to the life of the church in this world and a vision of the church in the world to come. They function as a grammar, not only as a system and structure but also as a manual articulating the rules and building blocks of a theology that is able to lead the Christian community to its *telos*: union with Christ.

This essay offers an overview of Augustine's Christocentric reading and preaching of the Psalms and is chiefly interested in the implications of this hermeneutic and homiletic for preaching today. A primary readership among homileticians and preachers is, therefore, intended. The discussion advances in the following way. The first section focuses on early Christian readings of the Psalms (as preserved in New Testament texts) to highlight salient aspects of the broader Christian tradition that Augustine inherited and from which he wrote and preached. The second explores Augustine's personal love of the Psalms, especially in the *Confessions*. The third analyzes excerpts from Augustine's sermons on the Psalms. The fourth considers the significance of Augustine's practice for preachers today.

## 2. The Psalms in the Early Church

For two thousand years, from the inception of the church, Christians have cherished the Psalms. This love was inherited from their Jewish siblings. One sees this from the sermons of Basil the Great in the fourth century to Dietrich Bonhoeffer's *Prayer Book of the Bible* in the twentieth century. Athanasius's *Letter to Marcellinus* serves as an early example of how closely knit the Psalms were with the hearts of the leaders of the early church. In this letter, Athanasius famously describes the Psalms as a "mirror of the soul" (Athanasius 1980, p. 24). The earliest evidence of Christian love for the Psalms appears in the New Testament documents themselves. New Testament authors cite the Psalms more than any other book in the Hebrew Bible.[2] Memorable New Testament phrases and concepts such as "there is no one who is righteous, not even one" (NRSVUE)[3] in Romans 3:10 and the priesthood of Jesus in Hebrews 5–7 find their origin in the Psalms. From the earliest days, when the first Christian communities sought to understand Jesus and themselves, they reached for the Psalms.

The Psalms were used in sermons just as they were in letters. According to Acts 2, Peter cites two different psalms (Pss 16 and 110) in his Pentecost sermon to make Christological claims. In Paul's first full-length sermon included in Acts (his in Acts 13 delivered in Pisidian Antioch, which mirrors Peter's in Acts 2), he quotes the Psalms at least three times (Pss 2, 16, and 89). The use of the Psalms in sermons is relevant to this study because it shows that the first Christians had no qualms with using the Psalms outside their original liturgical contexts. Additionally, the Psalms were often read as prophetic texts,[4] a practice that Luke traces back to Christ himself. According to Luke 24:44–45, Jesus infused the Psalms with Christological significance, stating, "'These are my words that I spoke to you while I was still with you—that everything written about me in the law of Moses, the prophets, *and the psalms* must be fulfilled.' Then he opened their minds to understand the scriptures" (emphasis added).[5] These examples demonstrate that the first Christian communities used the Psalms in preaching and teaching, with Christ at their center. But this is still a step removed from reading the Psalms as the very words of Christ.

In his essay, "Christ Prays the Psalms: Israel's Psalter as Matrix of Early Christology," Richard Hays takes up this matter. Hays homes in on Romans 15:3: "For Christ did not please himself, but, as it is written, 'The insults of those who insult you have fallen on me.'" The verse Paul quotes from is Psalm 69:9. The full verse states, "It is zeal for your house that has consumed me; the insults of those who insult you have fallen on me". According to Hays, Paul engages in a form of "Christological ventriloquism" by putting the words of the psalm on the lips of Christ (Hays 2005, p. 104). What is equally important is that "Paul does not seek to explain or justify his identification of the psalmist's first-person singular pronoun with the figure of Christ," indicating that "the Christological interpretation of this psalm must have been an established tradition in early Christianity before Paul's writing of the book of Romans" (Hays 2005, p. 102).

Hays offers support with other examples to demonstrate that Romans 15:3 is not an anomaly. A few verses later in Romans 15, Paul does this again: "For I tell you that Christ has become a servant of the circumcised on behalf of the truth of God in order that he might confirm the promises given to the ancestors and that the gentiles might glorify God for his mercy. As it is written, 'Therefore I will confess you among the gentiles and sing praises to your name'" (Rom. 15:8–9). Again, the words (this time from Ps. 18:49) are attributed to Christ by context and use of the first-person singular pronoun (c.f. Hays 2005, pp. 102–4).

Several other instances occur in which Christ speaks the Psalms, such as John 2:17, where the disciples witness their Lord's anger in the temple and recall a psalm: "Zeal for your house will consume me" (c.f. Ps. 69:9).[6] Once again, the psalm is attributed to Jesus with the first-person singular pronoun as though he had said these words.[7] But, the most significant instance of Christ praying the Psalms is at the cross when he prays Psalm 22:1: "At the ninth hour Jesus cried with a loud voice, 'Eloi, Eloi, lema sabachthani?' which means, 'My God, my God, why have you forsaken me?'" (Mark 15:34).[8]

From this survey, it is clear that early church theologians held the Psalms close to their hearts. Only a few of the citations from the Psalms that appear in the New Testament are mentioned above. These examples demonstrate that the first Christians engaged the Psalms in various practices and interpreted them in a variety of ways. Christians used the Psalms in worship as indicated by the apostle Paul's encouragement to the Ephesians to sing psalms (Eph 5:19) or the disciples incorporating a psalm into their prayer for boldness (Acts 4:24–30). But Christians also used them in preaching and teaching, both in speech and in writing. Christ's own statement in Luke 24:44–45 leads Christians to read the Psalms Christocentrically as though they are fulfilled in Christ. But the most important observation for the following discussion is that, from the earliest decades of the church's life, Christians have heard *Christ's very words* in the Psalms and not merely King David's or Asaph's words *about* Christ. Centuries later, this hermeneutical move is still being practiced by Augustine.

### 3. The Psalms in Augustine's Life

Augustine followed closely in the tradition of the New Testament writers and early Church Fathers. Augustine's love for the Psalms is clearly evident in his *Confessions*, likely written between 397 and 400. His spiritual autobiography reveals how he turned to the Psalms not as a preacher in search of a text,[9] but as a Christian in search of Christ.[10] In the *Confessions*, one sees that Augustine's love for the Psalms was not only public and pastoral but also personal and intimate. Augustine came to these texts not only for his flock but for himself. Augustine's autobiography takes the form of a prayer and address to God.[11] To paraphrase Rowan Williams, he recognized his own obscurity to himself and sought self-understanding through God's revelation.[12] At the center of Augustine's story is God.

Many have noted the prominence of the Psalms in the *Confessions*. "The very first sentence of Augustine's *Confessions* is a quotation from the Psalms, and for the rest of the work hardly a page goes by without at least one such reference," writes Williams. "It would not be an exaggeration to say that the narrative autobiographical voice of the *Confessions* is systematically blended with the voice of the psalmist" (Williams 2016, p. 25).

Peter Brown also notes the prominence of the Psalms in the *Confessions*. "It is not surprising," he writes, "that the *Confessions*, suffused as they are with a dramatic sense of God's interventions in Augustine's life, are studded with the language of the Psalms. This was, in itself, a startling literary innovation: for the first time, a work of self-conscious literary art had incorporated, (and most beautifully), the exotic jargon of the Christian communities" (Brown 2000, p. 168). The metaphor of "stud" may be understood in more than one way, but it would not be unfaithful to Brown's statement or Augustine's work to see the Psalms as supporting and strengthening Augustine's life story the way vertical, parallel beams support and strengthen ("stud") the walls of a home.[13]

In at least one place in the *Confessions*, Augustine himself draws attention to the role this Old Testament book played in his spiritual development. The passage is worth quoting at length.

> My God, how I cried to you when I read the Psalms of David, songs of faith, utterances of devotion which allow no pride of spirit to enter in! I was but a beginner in authentic love of you, a catechumen resting at a country villa of another catechumen, Alypius. My mother stayed near by us in the clothing of a woman but with a virile faith, an older woman's serenity, a mother's love, and a Christian devotion. How I cried to you in those Psalms, and how they kindled my love for you! I was fired by an enthusiasm to recite them, were it possible, to the entire world in protest against the pride of the human race. Yet they are being sung in all the world and 'there is none who can hide himself from your heat' (Ps. 18:7). (Augustine 1992, p. 160)

Augustine believed that the Psalms refuted his pride as well as kindled and fired his faith. One is reminded of the pivotal role that his mother Monica played in his life and faith. The Psalms are set alongside her in this passage as profoundly influential in the spiritual development of this theologian.[14]

Augustine also found himself drawn to the Psalms because they provided words for prayer, and he would much rather pray God's words to God than his own words to God. Augustine scholar Edmund Hill paraphrases a playful saying of Augustine's, something this preacher once told his congregation: God "only likes the sound of his own voice" (Hill 1954, p. 464). Augustine believed that his own words and inventions were inadequate for speaking of God, much less speaking to him; after all, "God is inexpressible," he writes in *De Doctrina Christiana*. Augustine wrestled with this dilemma, of human inability to speak of God and to God: "While nothing really worthy of God can be said about him" God nevertheless graciously deigns to accept "the homage of human voices" and wishes us to "praise him with our words" (Augustine 1996, pp. 108–9).[15]

Augustine believed that God accepts human words in prayer and praise, but he still thought it far better to rely on the words that God provides than to look into his own heart for something to say. Words (by themselves) fall short of adequately teaching or fully giving life to the soul; only words suffused by the Spirit of God can do that (namely, teach and give life).[16] Without God's aid, what basis was there for Augustine to trust language's efficacy in prayer, or his own heart's ability to communicate with God? In the *Confessions*, he writes: "Without you, what am I to myself but a guide to my own self-destruction?" (Augustine 1992, p. 52). To address God, therefore, he needed something more than his own words. The Psalms fit this niche wonderfully for, in them, the Christian finds words for God *given by God*. Henry Chadwick describes Augustine's rationale:

> The acute sense of the inadequacy of words explains why Augustine at the beginning of the *Confessions* experiences difficulty in finding any way of addressing God. . . . The answer to the question he finds in . . . scripture. . . . The Bible consists of words, human indeed but for the believing community a gift of God so that within the sign there is also a divine reality. . . . So Augustine can address God in the way the psalmist did. (Chadwick 1991, p. xxii)[17]

Augustine's restless heart (as he famously termed it at the start of the *Confessions*)[18] also predisposed him to these texts—he deeply appreciated their emotion and honesty. Michael Cameron writes: "Augustine's attraction to studying the often vehement prayers of the Psalms is hardly a surprise. His restless spirit entered easily into the Psalter and its words, which channeled the most primitive human desires, cravings, fears, sorrows, and joys into the search for God" (Cameron 2015a, p. 23). Rowan Williams adds that the biblical identity Augustine found was realized most fully in the Psalms, an identity that "speaks of longing, failure, betrayal, acceptance, travel and homecoming" (Williams 2016, p. 16). Few books in the Bible are as emotionally charged or license such soul-bearing transparency. This was one more reason why Augustine loved the Psalms; it was one of the most *human* books of the canon, giving vent to the humblest and rawest emotions of the heart.[19] This dimension of the Psalms' witness was important for Augustine's understanding of the Incarnation. With reference to the language of lament in Psalm 88, Patout Burns has argued: "Augustine affirmed that in voluntarily assuming mortal human flesh, Christ had also taken upon himself not only bodily mortality but the actual sadness and fear that the human soul experienced in sympathy with its body when facing danger or suffering" (Burns 2022, p. 205).

These are but a few of the reasons Augustine loved the Psalms: their call to humility, their being provided by God, and Augustine's own restless heart. One may add to these their beauty as verse. Augustine was trained in classical rhetoric, so the unrefined nature of much of Scripture would have been somewhat off-putting, especially prior to his conversion. According to Carol Harrison, "Augustine is at pains to demonstrate that the Christian Scriptures can be analyzed according to the rules of classical eloquence and that they are not found wanting in this respect" (Harrison 2000, p. 217). Perhaps the Psalms, with their vivid metaphors and elegant parallelisms, were an oasis within the Bible for a lover of Cicero, Quintilian, Plato, and Plotinus.[20] Although Augustine loved the Psalms for a number of reasons, no reasoning can do justice to his love of the Psalms that does not begin and end

with Christ.[21] Augustine loved the Psalms because he loved Christ, and because he heard Christ's voice in them.

### 4. The Psalms in Augustine's Sermons

Augustine's lengthiest work is twice as long as *De civitate Dei*, a fact difficult to fathom for those who have read *The City of God*. His longest work is *Enarrationes in Psalmos*, a series of expositions (or a "running commentary") on the Psalms.[22] This anthology is "the most extensive treatment of the Psalter that has survived from the ancient church" (Cameron 2012, p. 166). Augustine took twenty-five years to complete the set, in which he offers at least one preachable exposition of each of the 150 psalms.[23] One might suppose *prima facie* that this homiletical production exhaustively comprises Augustine's preaching on the Psalms, but this would be incorrect for a number of reasons. Within the expositions of each of the psalms, there are often numerous references and allusions to other psalms outside the one under consideration. Also, Augustine's sermons outside the *Enarrationes* make constant reference to the Psalms.[24] The Psalms were more than preachable texts to Augustine: they were an underlying theological grammar that made all his sermons coherent and intelligible.

In Augustine's church, the Psalms were kept front and center in the liturgy. While no sacramentary from the church in North Africa survives, scholars have gathered enough clues from Augustine's works to reconstruct the typical order of worship. After the Scripture reading from the Old Testament or Paul, Augustine would send "word to the cantor which Psalm he wanted sung. The Psalm was sung in responsorial style, the cantor intoning the verses with the congregation joining in the refrain" (Harmless 2010, p. 124). The Psalms were part of every worship assembly and often were sung just before Augustine's sermon. They were fresh on his mind when he sat before his congregation to preach. Augustine did not preach from manuscripts. Rather he prepared through "prayer and study," often improvising as the occasion demanded. His sermons were preserved by *notarii* (essentially stenographers) who took notes of the sermons using shorthand (Harmless 2010, p. 124).

The homily presented Augustine with a special opportunity to demonstrate before his congregation scriptural exegesis, and, in a Christologically significant way, to bring the text to life. François Dreyfus discusses the liturgy in the ancient world, and calls the homily the privileged place of the actualization of the Scriptures: "La liturgie a toujours été le lieu privilégié de l'actualisation de l'Ecriture, et dans le judaïsme et dans la tradition chrétienne" (Dreyfus 1979, p. 23). Actualizing a biblical text in the homily means far more than sharing information about the text. It means corporate involvement in that text as well as creating a space for the living Word of God to speak.

Thus, Augustine preached the Psalms Christocentrically,[25] and, what is more, as the very words of Christ, using a hermeneutical strategy known as prosopological exegesis. In Hebrew and Greek texts from antiquity (such as poems, plays, and philosophical dialogues), speech was not marked by quotation marks, so the question "Who is speaking?" would have been a necessary one, even for the Psalms (Harmless 2010, p. 193). Michael Cameron writes about the hermeneutic of voice discernment:

> Augustine's tool for distinguishing Christ's voice behind the other voices in the text came from adapting a standard practice of his grammatical and rhetorical training called "prosopological exegesis" ... [which] refers to the work of literary analysis that identifies the various speaking voices of a poetic text. The speaking person, *prosopon* in Greek (literally, "face," which Latin writers often translated as *persona*, "person"), had to be identified ... without cues, line breaks, punctuation or even spaces between letters. (Cameron 2015b, p. 208)[26]

One observes Augustine's prosopological exegesis in various sermons. For example, we find it in his exposition of Psalm 57:1: "*Have mercy on me, O God, have mercy, for my soul trusts in you*. Christ is praying in his passion, *Have mercy on me, O God*. God is saying to God, *Have mercy on me*" (Augustine 2015b, p. 163).[27] In his second exposition of Psalm 31:1–11,[28] Augustine proclaims:

> Christ is speaking here in the prophet; no, I would dare go further and say simply, Christ is speaking. He is going to say certain things in this psalm that we might think inappropriate to Christ, to the excellent dignity of our Head, and especially to the Word who was God with God in the beginning. Some of the things said here may not even seem suitable for him in the form of a servant, that form which he took from the Virgin; and yet it is Christ who is speaking, because in the members of Christ there is Christ. I want you to understand that Head and body together are called one Christ. . . . Let Christ speak, then, because in Christ the Church speaks, and in the Church Christ speaks, and the body speaks in the Head, and the Head in the body. (Augustine 2015a, pp. 213–14)

In Psalm 57:3, Augustine hears in the psalmist's joy at divine deliverance "the prayer of Christ, as a man of flesh, praising God for rescuing and raising him up" (Burns 2022, p. 207). Augustine used this hermeneutical approach to discern which words in the Psalms were to be attributed directly to Christ. This was no simple matter. Patout Burns explains that Augustine "had to distinguish not only different modes of reference—particularly the historical and prophetic or typological—but also three different senses in which Christ was the subject or object of the biblical speech," as follows: "First, Christ could speak and be spoken of as the Word of God"; "second, the scripture refers to Christ as the Word of God incarnate," not only in such texts as the Johannine Prologue but also in Psalm 2:7; and "third, scripture can refer to 'the whole Christ in the fullness of the church'" understood as Christ's Body (Burns 2022, p. 15).

A subsequent tool that Augustine employs is prosopopoeia. If prosopological exegesis identifies the voice of the speaking Christ in the psalm, prosopopoeia imitates that voice. "Prosopopoeia" literally means "face-making". For Augustine, the words of the Psalms were not originally or primarily the psalmist's but those of Christ, the eternal Word. Only because these words flow from Christ can they be the psalmist's, the church's, and Augustine's. This is crucial for Augustine, not only that the Psalms offer the Christian the voice of Christ, but that the Psalms offer the Christian the Christian's own words. Rowan Williams writes: "The words of the Psalter are, as Scripture, the utterance of God: yet, they are simultaneously words addressed *to* God" (Williams 2011, p. 29). He continues:

> [In the *Enarrationes*, Augustine employs] an incarnational hermeneutic: God the Son in assuming humanity assumes all that humanity says to God, making his own even the cries of pain and doubt uttered by humans, so as to show that the transforming grace of God can work in situations of the gravest human extremity. Revelation in the words of Scripture does not come simply in the words we consider edifying and positive. And because the Son is eternally turned towards the Father in adoration and self-offering, what humans say can be taken up in that movement towards the Source of all, can become part of what the Son "says" to the Father and so brought into the reality which alone can heal and unite the fractured voices of creation. (Williams 2011, p. 30)

The voice of Christ in the Psalms mirrors Christ's incarnation. "The Psalms represent the unifying of the divine and human voice in Christ" (Williams 2016, p. 27).

Two theological concepts capture Augustine's Christology and soteriology: *kenosis* and deification. On the one hand, in becoming human, Christ emptied himself, as Paul says in Philippians 2:6–8: "Though he existed in the form of God, [Christ] did not regard equality with God as something to be grasped, but *emptied* [ἐκένωσεν, from κενόω, "to empty"] himself, taking the form of a slave, assuming human likeness. And being found in appearance as a human, he humbled himself and became obedient to the point of death—even death on a cross" (emphasis added). The incarnational self-emptying of Christ is a central concept in Augustine's thinking as well as a theme that dominates his preaching on the Psalms.[29]

Deification is another key concept for Augustine,[30] for it is by virtue of Christ's descent to humanity that humanity is able to ascend to God to be "participants of the divine nature" (2 Pet 1:4). In Christ, Christians become united with God. This is an important aspect of

Augustine's soteriology, and it has strong ecclesiological implications, that the church is to be transformed into the glory of the Son. Augustine preaches:

> Let us now listen to the Lord Jesus Christ speaking in our psalm's prophetic words and remember that, though the psalms were sung long before the Lord was born of Mary, they were not sung before he was Lord. . . . That divine person, equal to the Father, became a sharer in our mortality, a mortality that belonged not to him but to us, so that we might share the divine nature that belongs not to us but to him. (Augustine 2015c, p. 181)

Evident in this passage is the claim that the Psalms are the Lord's and not merely the psalmist's. Also, evident here is Augustine's understanding of Christ's salvific work of descent into creation to raise it back up to God.

Jason Byassee captures the *kenosis*-deification dynamic when he writes: "The Psalter's work on its readers is a reflection, or even extension, of the work of Christ. As Augustine insists each time he lays out his Christological hermeneutic for the psalter, Christ accepts our human words, and then gives us divine ones of praise in return" (Byassee 2007, p. 202). Thus, in Christ, prayer can truly become "God's breath in man returning to his birth" and "heaven in ordinaire," to borrow a couple of phrases from the poet George Herbert (Herbert 1961, p. 44). This is the great reversal enacted and effected by the work of Christ, which makes prayer and salvation possible. Cameron writes: "By joining them to his person and to his cross, [Christ] retroactively *transforms* them. When Christ transposes all humanity into himself on the cross, he trades humanity's death for divine life" (Cameron 2012, p. 208). If this is so, Byassee is correct in describing the preached word's sacramental efficacy. "The Psalter functions in a way analogous to Christ. It not only depicts, but also *effects* the divine *kenosis* and human *theosis* that is at the heart of scripture" (Byassee 2007, p. 203).

Seen this way, the Psalms serve as an indispensable key to a Christian's growing in the likeness of Christ. The Psalms offer a grammar, a formational training, in which the human person is transformed by Christ. As songs, the Psalms communicate on various levels, uniquely fitted for embodied education. Harrison observes, "the Fathers thought that the Psalms, and in particular, hearing or singing the Psalms, could enchant the soul, unifying, harmonizing, and ordering it so that it converted towards, participated in, and resonated with the cosmic harmony of the Creator" (Harrison 2011, p. 207). A book of prayers and songs is uniquely suited to do this sort of work in the hearts and souls of human beings: to transform their minds and desires into the likeness of Christ.

Another theological theme that appears in the sermons is the *totus Christus*. Augustine is the first to use this specific term, which means "the complete Christ" (Williams 2018, p. 74), and most likely drew the hermeneutical principle from Tyconius's *Book of Rules* (Harmless 2010, p. 158). This concept became the "gravitational center of his exegesis of the Psalms" (Harmless 2010, p. 158). It refers not simply to the two natures of Christ in one person but to Christ and his community.[31] Augustine claims,

> Christ speaks in this psalm. Many things have been said in the name of his body, but the head is speaking too, though not in the sense that they are distinct from each other like two persons: now the head and now the body. To distinguish them like that would be to divide them, and then they would not be two in one flesh. But if they are two in one flesh, do not be surprised if the two speak with one voice. (Augustine 2015c, p. 197)

The *totus Christus* is the head (Christ) and the body (the church) and was an important aspect of Augustine's doctrine of the incarnation.[32]

*Totus Christus* theology is part of Augustine's larger theology and corpus of writings,[33] and it appears throughout the sermons. Commenting on the words of Christ to Saul on the road to Damascus (in Acts 9:4), Augustine claims that when someone steps on another person's foot, the mouth will not simply say, "Why are you stepping on *my foot*?" but also "Why are you stepping on *me*?" Christ so joins himself with his people that their persecution becomes his persecution, and their prayers become his prayers (Augustine 2015a, p. 213).

Then, in his exposition of Psalm 142, Augustine once more references Genesis 2:24 and Ephesians 5:32:

> Christ and the Church, two in one flesh. The fact that they are *two* points to the distance between us and the majesty of God. They are two, undeniably, for we are not the Word, we were not with God in the beginning, not through us were all things made. But when we consider the flesh, there we find Christ, and in Christ we find both him and ourselves. Small wonder that we find this mystery in the psalms. There he says many things in his own name as head and many others in the name of his members, yet all of it is said as though one single individual were speaking. Wonder not that there are two with one voice, if there are two in one flesh. (Augustine 2015d, p. 374)

This passage shows that Augustine never lost sight of the distinction between Christ and the Christian community. Christ and the church "are two, undeniably, for we are not the Word, we were not with God in the beginning, not through us were all things made". In an important sense, Christ is Christ (*head and body*), and the church is not. Yet, Augustine also argues that when Christians consider Christ in the flesh, there they find not only Christ but "both him and ourselves".

By the grace of God, human beings are invited to partake in the life of Christ. By the power of the Spirit, Christians are incorporated into the person of Christ. This new identity as the body of Christ is a gift. As with all gifts God gives, it nevertheless must be understood, embraced, and grown into. This is where preaching that aims at formation is necessary, in helping the church to become what God says it is: the body of Christ.

## 5. The Psalms, Preaching, and Christian Formation

Up to this point, this essay has maintained a historical focus, drawing attention to the ways that the earliest Christian communities understood the Psalms Christocentrically. Some preachers, such as Augustine, went beyond a Christocentric reading to read the Psalms as though Christ himself was speaking in them. Many excerpts in Augustine's sermons demonstrate this conviction. Moreover, in the Psalms, the body of Christ can hear its own words. In praying them, the church grows up into Christ. In leading a congregation to the Psalms, the preacher helps the people of God hear the voice of Christ, learn his language, and become like him.

What remains is to discuss the significance of this for preaching today. Augustine's interpretation and preaching of the Psalms offer preachers at least two lessons about the work of preaching. First, as texts, the Psalms offer the church stepping stones in the journey toward Christ.[34] There is room for conversation here about appropriate and inappropriate ways to read the Psalms, whether and to what extent Christians should amplify the voices of psalmists with the voice resonances of Christ speaking. Even so, the preacher who listens for Christ's voice in the Psalms stands in a long tradition of saints who have read them in this manner. What if preachers listened with Augustine for the voice of Christ in the Psalms, the Christ who stands in solidarity with his church and who unites the church to God?

Second, Augustine teaches preachers about the true *telos* toward which all preaching is aimed, and, for that matter, the *telos* toward which all Christian ministry is aimed. This *telos* is incorporation into the divine life, the union of God and God's people in Christ. This is deification, whereby created human beings participate in the divine life made possible through Christ who became human, pouring out his own life to raise the church back up to God. Preachers may work with Augustine in preaching the Psalms, laboring to remind the church of Christ's voice, which is the congregation's voice also.[35] What Augustine predicated of Psalm 22 applies more broadly to his understanding of the Psalms as giving voice to all believers: Augustine "invited his hearers to identify themselves in the psalm's prayer of repentance and plea for deliverance that their head had offered, not for his own failures and abandonment, but in solidarity with them" (Burns 2022, p. 276). Contemporary preachers can make audible for their listeners the theologically rich strains of Augustine's

Christological understanding of the church's speech in liturgy and beyond. To borrow Augustine's words preached over the Eucharist, "Be what you can see, and receive what you are" (Augustine 2007, p. 318). The preacher who leads the church to the voice of Christ in the Psalms tells the congregation, "Be what you can hear, and receive what you are".

Talk of the church's union with Christ may set off some alarm bells. There is danger in collapsing the distinctions between Christian and Christ. Jerusha Matsen Neal warns against the "dangerous preacher," one whose preaching becomes the word of God, one who herself or himself "*becomes* the Word's flesh, eclipsing the particularity of Jesus" (Neal 2020, p. 44). It is crucial to keep in mind that in the book of Acts, the church never succeeds or replaces Jesus. Jesus is ascended yet still active in the story of the church (see Neal 2020, pp. 58ff; see also Van Driel 2020, pp. 10ff). Even with his emphasis on the church as the body of Christ, Augustine does not fail to distinguish between the two.

As essential as it may be to remind the preacher (as well as each congregant) that "God is God and you are not," one wonders, what then? What do preachers have to say after clarifying human otherness and estrangement from God? Christians affirm the categorical difference between Creator and creation (as Augustine did) and ought to be transparent about the ways in which Christians chronically fail to live up to their calling. Yet, one wonders if there is a subsequent word. After these truths are articulated and accepted, do preachers have any other message to deliver?—and not simply a pacifying message admonishing the church to wait for God to overcome all present ailments in the life to come. Is there a word for *today*, for the here and the now? Is there a word for the people of God that articulates the hope of growth into the likeness of Jesus Christ?

There is need and opportunity for preachers to embrace Augustine's Christocentric homiletic. Sermons have many functions, to be sure, and wise pastors discern what their congregations need any given Sunday—be it leadership, or a sort of therapy, the teaching of biblical truth, or the proclamation of the gospel. Nevertheless, sermons that aim at Christological formation are, perhaps, more rare than they ought to be.[36] Christians are nourished by the reminder that their lives are "hidden with Christ in God" (Col 3:3). This new location and identity *in Christ* is a gift as well as a summons: "to grow up in every way into him who is the head, into Christ" (Eph 4:14). To borrow an image from New Testament scholar N. T. Wright, Christians are given a "suit of clothes designed for us to wear in our full maturity. And most of us, putting the suit on week by week, have to acknowledge that it's still a bit big for us, that we still have some growing to do before it'll fit" (Wright 1996, p. 12). Sermons serve the purpose of helping the church grow into its rightful maturity, to imitate Christ, and to participate in his life.

The corporate aspects of this project cannot be forgotten. "We must acknowledge that we are members of a 'we'," Charles Pinches writes. "The sermon is a place where the 'we' is particularly evident" (Pinches 2006, p. 174). The "we" here is not simply *such and such congregation* or *these particular dozens of persons who form this community*, but nothing less than *the body of Christ*. We need to be reminded, Jürgen Moltmann contests, not simply about "*who we are* but also *where we belong*" (Motlmann 1993, p. 286). Who we are is constituted by Christ and created by his word. Barth reminds the preacher that the Word of God [in its full, threefold sense] makes "proclamation proclamation and . . . the Church the Church" (Barth 2010, p. 85). Whether the sermon makes this explicit or not, the church has no present being or future fulfillment apart from Christ. These theological bearings offer preachers a richer and larger purpose, not simply of this or that sermon, but of preaching itself.

Augustine seemed to believe that, in Christ, the church is more than it seems to be. He believed that through preaching, the church grows. The church grows to "maturity, to the measure of the full stature of Christ" (Eph 4:13). If preaching aims at this *telos* (i.e., to form Christian individuals and communities into the image of Christ), then the Psalms may be an invaluable resource to the preacher. The Psalms are a grammar of the Christian life, making the struggles and victories of all it means to be human intelligible. The Psalms are a grammar, a primer, so that in learning these words, the church learns the words of Christ, and learns its own self. Preachers can preach the Psalms with Augustine. They can

remind their people that Christ is with them, and that through Christ, the church is raised up to God.

**Funding:** This research received no external funding.

**Institutional Review Board Statement:** This research did not require institutional review.

**Informed Consent Statement:** This research did not require informed consent.

**Data Availability Statement:** No new data were created or analyzed in this study. Data sharing is not applicable to this article.

**Conflicts of Interest:** The author declares no conflict of interest.

## Notes

1    Biblical scholar W. H. Bellinger makes use of this metaphor in relation to the Psalms (Bellinger 2019).

2    The only serious rival to the Psalms is the book of Isaiah, Bellinger claims (Bellinger 2019, p. 2).

3    All Scripture quotations that are not embedded within another quotation are taken from the NRSVUE (National Council of the Churches of Christ in the United States of America 2022).

4    Reading *as prohphetic* typically non-prophetic biblical material became more prominent in the Second Temple period. One sees this in writings from Qumran. For instance, see the *Rule of the Community* (1QS) 8:13–16a, which references Isaiah 40:3 and reads, "... the expounding of the Law, decreed by God through Moses for obedience, that being defined by what has been revealed for each age, and by what the prophets have revealed by His holy spirit" (Parry and Tov 2014, 1 p. 25). More directly related to the Psalms is 11Q5 27:2: "Now David the son of Jesse was wise and shone like the light of the sun, a scribe and man of discernment, blameless in all his ways before God and men. The LORD gave him a brilliant and discerning spirit, so that he wrote: psalms, three thousand six hundred; songs to sing before the altar accompanying the daily perpetual burnt offering, for all the days of the year, three hundred and sixty-four; for the Sabbath offerings, fifty-two songs; and for the New Moon offerings, all the festival days, and the Day of Atonement, thirty songs. The total of all the songs that he composed was four hundred and forty-six, not including four songs for charming the demon-possessed with music. The sum total of everything, psalms and songs, was four thousand and fifty. All these he composed through prophecy given him by the Most High" (Wise et al. 1996, p. 452). The point here is that texts like the Psalms were being considered prophetic by Jewish communities even before the first century.

5    Luke departs from a standard tripartite formula, "the Law, the Prophets, and the Writings," transposing "the psalms" in place of "the Writings," which "signals the special significance of the Psalter as a medium of revelation of Jesus's identity" (Hays 2016, p. 234).

6    See also Heb 2:12 and 10:5–7.

7    For more on the significance of this, see (Hays 2005, p. 105; Dodd 1952, pp. 57–60).

8    The significance of this was not lost on Augustine. As Michael Cameron observes, "It is impossible to overstate the importance of [Psalm 22] for Augustine" (Cameron 2015b, p. 217).

9    A somewhat conventional practice. For example, Alan of Lille noted seven centuries later that the Psalms were good texts to preach (Alan of Lille 1981, p. 20).

10    Possidius relates that Augustine spent his last days weeping over his sins and praying the Psalms: "He had ordered that the four psalms of David that deal with penance to be copied out. From his sick-bed, he could see these sheets of paper every day, hanging on his walls, and would read them, crying constantly and deeply" (Possidius as quoted in Brown 2000, p. 436).

11    "Purely formally, the whole of the *Confessions* is a prayer; to work out who I am, I need to be speaking to and listening to God" (Williams 2016, p. 3).

12    "Once we have recognized how obscure we are to ourselves we somehow see that only in relation to the infinity of God can we get any purchase on the sort of beings we are" (Williams 2016, p. 4).

13    Henry Chadwick makes a statement similar to Brown's: "Citations from the Psalms are even made integral to the literary structure of the work, so that in several cases a citation links the books together like a coupling" (Chadwick 1991, p. xxii).

14    It is not surprising—though not without a touch of irony—to see Augustine quoting a psalm in stride in the middle of this passage about the role the Psalms have played in his life!

15    Regarding the givenness of God's word and the reliance on the given word of God to speak for and to God, preachers may be reminded of the words of Karl Barth, that "preaching must conform to revelation" and "the point of the event of preaching is God's own speaking (*Deus loquitur*)" (Barth 1991, p. 47). Without God's words, what do preachers have to say?

16    Cf. Cary (1997). One may hear an echo here (regarding Augustine's skepticism of the efficacy of words) and 2 Corinthians 3:6, a verse that was pivotal in Augustine's thinking: "The letter kills, but the Spirit gives life".

17    One also hears the theology and writing of Bonhoeffer: "Therefore we must learn to pray. The child learns to speak because the parent speaks to the child. The child learns the language of the parent. So we learn to speak to God because God has spoken and

speaks to us. In the language of the Father in heaven, God's children learn to speak with God. Repeating God's own words, we begin to pray to God. We ought to speak to God, and God wishes to hear us, not in the false and confused language of our heart but in the clear and pure language that God has spoken to us in Jesus Christ" (Bonhoeffer 1996, p. 156).

18    "You have made us for yourself, and our heart is restless until it rests in you" (Augustine 1992, p. 3).

19    William Harmless also makes a connection between the Psalms and affect: "The Psalms provided a paradigm for the affections, offering a path both of purgation and of delight that instilled in the faithful the graced courage to make the often grueling journey to God" (Harmless 2010, p. 196).

20    David Wilhite briefly discusses the influence of Cicero and the Neoplatonists on Augustine. See (Wilhite 2017, p. 242). In another place, Wilhite remarks that "virtually all of the surviving sermonic material [from the pre-Nicene period] owe a big debt to the classical rhetoricians, such as Aristotle, Cicero, and Quintilian". He follows with an illustration of Augustine's dependence on these sources (Wilhite 2019, pp. 55–56).

21    Augustine "read the psalms . . . primarily in relationship to Christ" (Brown 2014, pp. 8–9). O. C. Edwards notes, "What he found [in the Psalms] was very different from the intention of the psalmist, but it is certainly consistent with the faith of the New Testament. What he came up with is the sort of thing one immersed in the Christian faith and spirituality is likely to think when reading a psalm" (Edwards 2004, p. 115).

22    Augustine's sermons do not typically enjoy the attention his other writings receive. As one scholar said, Augustine's Bible is "primarily the Bible of a preacher" (La Bonnardière 1999, p. 212). Augustine's preaching seems to be gaining attention in recent years. For a few notable examples, see (Harmless 2010; Kolbet 2010; Sanlon 2014; Burns 2022; and Glowasky 2020).

23    It is not known whether Augustine actually preached every one of these expositions himself. For a brief overview of the *Enarrationes in Psalmos*, see (Old 1998, pp. 358–60).

24    As just one example (chosen arbitrarily), Augustine's short Christmas sermon (sermo CLXXXVII) contains at least three references to the Psalms. See (Augustine 2002).

25    Of those Church Fathers who commentated on the Psalms, Augustine is the "most consistently christological" (Price 2011, p. 14).

26    See also (Cameron 2012, p. 171). Cameron mentions Drobner, who has conducted more thorough investigations of this practice. See (Drobner 1990, pp. 49–63).

27    Augustine's Psalm 56:2. In the Septuagint, Psalms 9 and 10 constitute a single psalm; the LXX numbering of Psalm 10 through Psalm 147 is one behind the numbering in translations based on the Hebrew text.

28    Augustine's Psalm 30:1–11.

29    For instance, he says in one place, "It was by humility that [Christ] opened a way for us. . . . God himself became humble". (Augustine 2015a, p. 117).

30    The logic of deification is often present (such as in these sermons) whether it is ever named as such. It is true, in fact, that Augustine seldom deploys the term "deification". Robert Puchniak suggests a reason for this, drawing on *City of God*. Augustine writes in a pagan context in which human beings were all too eager to deify themselves. The result would be that "false gods" are worshipped while the "'living god' is neglected". Augustine hesitates to use the term lest he be misunderstood (Puchniak 2006, p. 131). For the concept of deification in Augustine, see (Meconi 2008; 2013, pp. 135–174; 2014).

31    Pauline passages stand behind this concept, such as Eph 1:22–23; 5:22ff; Col 1:18; 2:18; and 1 Cor 12:12ff.

32    "The human collectivity that is the Church is one *homo* with Jesus, and so fully one with the Word" (Williams 2018, p. 74).

33    Burns emphasizes this aspect of Augustine's thought: "As human, Christ had become the head of a social body, the church, in which the baptized were joined to him and to one another. The sin and weakness of the members were symbolized in the mortality and the suffering of Christ, their head. The immortality and new life of the head was realized in the moral renewal of the members and, eventually, their bodily resurrection. As Christ really died and rose in his own flesh, he symbolically died and rose in the baptism of his members. His bodily resurection was realized in their spirits: their minds were lifted up to him; their hearts were moved to love God and neighbor; they lived in expectation of joining him physically" (Burns 2012, p. 435).

34    For aid in preaching the Psalms, see (Fleer and Bland 2005; Langley 2021; Long 2014, pp. 557–68; Long 1989, esp. pp. 43–52; and McCann and Howell 2001).

35    Annette Brownlee underscores that the task of preaching concerns discerning the body and helping congregants understand the Christ-given reality they inhabit as Christians. "Preaching is a practice in which we are called to discern the body as a way of life. Preaching, as Augustine described it, is an 'audible sign' with bread and wine as 'visible words.' In preaching we have the privilege of naming and describing the reality already present in the church as it gathers in the name of the risen Christ and breaks bread together. It is our privilege to describe the way of life springing from this reality" (Brownlee 2018, p. 101; see also Jenson 1978, pp. 4–5; Fitzgerald 1999, p. 744).

36    An anecdotal example of the problem is seen in the work of Sally Brown and Luke Powery, *Ways of the Word*, which helpfully includes an essay on "Preaching and Christian Formation". Brown delineates three stages of development: from discovering or rediscovering the Christian faith to becoming a true disciple, and finally to learning to live out one's faith (Brown and Powery 2016, p. 238). These phases are fine, insofar as they go; they accurately describe levels of growth. Yet, it is possible to conceive of these sorts of thresholds in individualistic ways which leave the disciple unincorporated in Christ's body. The *possibility* of

Christian formation often fails to be theologically grounded in the person and work of Christ or aimed at the consummation of Christian history, the "gather[ing] up [of] all things in [Christ], things in heaven and things on earth" (Eph 1:10).

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
