# Peer review of "“Christ Is Speaking”: The Psalms as the Grammar of Augustine’s Sermons"

_religions, doi:10.3390/rel15040414_

Round 1
Reviewer 1 Report
Comments and Suggestions for Authors
The article provides an overview of Augustine's reading of the Psalms. It especially focuses on Augustine's incarnatorial hermeneutic.
A major lack of this article is that no distinct or original thesis is developed. The aim of the article remains unclear.
Moreover, the article often focuses more on the presentation of scholarship than on Augustine's own voice.
There are several comments to specific arguments / remarks:
- Chp. 2 presents the (christological) use of the Psalms in the New Testament. It is shown that the New Testamental authors regarded the Psalms (or some Psalms?) as representing Christ's own words. This interpretation is exciting. The antithesis of Christ's very words and words about Christ is maybe one-sided (cf. Hebr 1-2). Moreover, the author's claim to present the tradition of Augustine is somehow misleading because the Christian tradition of exegesis of the Psalms is omitted (for example, Origen, Ambrose). The argument that "the 1st century Church made the same hermeneutical move that Augustine would make" is a superficial statement; cf. moreover p. 8 l. 390-393 where the author emphasis that Augustine's approach (highly) differs from the christocentric exegesis of Psalms in the New Testament. The author itself refers to Augustine's differentiate hermeneutic of the Psalms (which is not found in the New Testament; moreover, within the New Testament itself different approaches to the Psalms should be distinguished).
- The idea of kenosis in Augustine's work is well studied (cf. for example Kantzer Komline in Paul T. Nimmo and Keith L. Johnson, Kenosis : the self-emptying of Christ in scripture and theology). However, I would deny that the term "theosis" (in a strict sense) is appropriate for Augustine's soteriology. Augustine taught the idea of participation in Christ, however he clearly differentiated between God and (saved) humans. Moreover, he did not use terms of theosis (such as Greek Church fathers [Gregory of Nazianz, Athanasius]). Thus, the opposition of kenosis-theosis in Augustine is misleading and not precise. If the author believes that Augustine understood his soteriology as theosis he has to substantiate this.
- Augustine explicitly and often emphasises the difference between Christ and Christians (cf. p. 9). This should be also considered as important hermeneutic axiom.
- Moreover, the totus Christus theory is connected to Augustine's doctrine of incarnation.
- Fiedrowicz's analysis of the exegesis of the Psalms could be very helpful: Fiedrowicz, Psalmus vox totius Christi : Studien zu Augustins "Enarrationes in psalmos"
- While quoting Augustine's work it would be helpful to refer not only to the English translations (i.e. p. 3: besides Augustine 1992, p. 160, also a reference to book and chapter of the Confessions).
The historical overview of Augustine's exegesis of Psalms (chp. 3-4) is merely a summary of the present scholarship. Moreover, the article uses some categories (such as theosis or Christocentric exegesis) quite superficial; a more explicit analysis of Augustine's own words would be helpful.
If the main interest of the article is the "significance of [Augustine's exegesis of Psalms] for preaching today" (l. 398) the article should emphasise this more strongly. This approach could be very exciting.
Author Response
I want to thank you for such a careful reading of my paper, and for bringing to bear upon it an obvious expertise in Augustine studies. As you suggest, I have tried to make the intent of my article much more clear. I have done this in a few ways: first by adding a line or two in the introduction about the intent of the article and my assumed readership. Second, I have also added a number of sources relative to preaching, and elongated the conclusion of the paper which focuses on homiletical takeaways in order to put more weight there, on preaching. My intent certainly was not/is not to advance the field of Augustine studies (I have no illusions of doing that), but rather to acquaint homileticians with an aspect of Augustine's preaching with which most non-specialists and preachers would be unfamiliar. I believe that this endeavor makes the work worthwhile and beneficial, mostly for preachers.
I appreciate the point about failing to represent any sort of tradition: I hop from the New Testament to Augustine, without explicating much else of a tradition. I have tried to adjust my language to suit this lack of coverage.
An obvious adjustment is in the choice of deification, a more palatable word than theosis, which conjures up an Eastern provenance. I have supplemented that section with secondary sources, and answered anticipated pushback about Augustine's infrequent use of the word deification. Regardless of whether the term is used, the logic of the term is assumed, or so I see.
I have also inserted clarifications regarding the difference between Christ and church in order to avoid collapsing and conflating the two entities.
As to the issue of superficiality, I hope that I have answered that above. Before reading your review, I had a sense of this weakness. Nevertheless, the article is confined to a word count, and meant to be brief. I am attempting to introduce preachers and homileticians to an aspect of historical preaching that they otherwise may not engage. I see the benefit of the work in the integration of the then and the now of preaching, and the exploration of the possibilities of preachers preaching the Psalms, and also finding Christ there--the human person and the divine Son--and with him to find ourselves, too. I find these concepts/this theology in Augustine beautiful and inspiring and hope that others will, too.
Again, let me express my sincere gratitude for the level at which you have engaged my work, for your criticisms and suggestions that have, I hope, made the piece stronger in your eyes.
Reviewer 2 Report
Comments and Suggestions for Authors
It is hard to judge this paper as it seems to be for a general homiletics issue. The essay is standard and very general. There is a limited and narrow engagement with scholarship on Augustine. However, as it is for a homiletics issue, this essay may be suitable.
The author should explain how these sermons were compiled. Does the author think that Augustine compiled these separating them from his other sermons? Does the author think that these were dictated (as many of Augustine's writings) or that these are notes from homilies by a stenographer in attendance (as some homilies)? There are many presumptions made by the author about the Enarrationes that need more study.
Who does the author think that the audience for these reflections on the Psalms was? Augustine's congregation, his monastic community, reflections for other bishops?
What about Ambrose's use of the Psalms? What about Manichaean Psalms which we know were influential for Augustine?
The essay has a tendency to overstate and has a limited engagement with Augustine's theology, especially of the totum corpus.
line 111 - "Nowhere is Augustine's love for the Psalms..." This can be restated without 'nowhere...more...than' to be more convincing.
line 118-119 - "Paradoxically, it is God (not Augustine) who is the central character of Augustine's story." - Why is this? It is prima facie false. If God is a character, then both Augustine and God are central. Unless the register used here is mixed.
line 146 - The reference to Monica is a non sequitur in the essay.
line 160-161 - This sentence is a false dichotomy. The author seems to think that 'human words' and 'God's words' are in opposition, both in prayer and in thought. This shows a fairly basic - and misguided - understanding of Augustine's 'theory' of language.
line 166-167 - Repeats the above error. "His words" vs "God's words".
line 187 - "vent" - Augustine does not hold the 'carthatic view' of emotions. This is said about the Psalms, but also 'why' Augustine loved them. The author has a superficial presentation of the emotions in Augustine.
line 197 - "Would have been" instead "was" - Augustine says this himself, it does not need to be couched in such terms.
line 202 - I am curious if the author can point to explicit references to Quintilian in Augustine. References to Plotinus and Quintilian are at best stipulated in Augustine's writings. Cicero, Seneca, Plato, Porphyry, etc. and directly cited.
line 207 - the Enarrationes are not a work. The author should explain how these sermons were compiled. Does the author think that Augustine compiled these separating them from his other sermons? Does the author think that these were dictated (as many of Augustine's writings) or that these are notes from homilies by a stenographer in attendance (as some homilies)? There are many presumptions made by the author about the Enarrationes that need more study.
line 212 - why "preachable" when the footnote acknowledges that it is not known if these writings were preached. - Also, who does the author think that the audience for these reflections on the Psalms was? Augustine's congregation, his monastic community, reflections for other bishops?
line 267 - the block quotation that ends here has no explanation. It might also be better used if broken - for the last two sentence would support the inchoate section on the Totus Christus.
line 314 - "This is Augustine's soteriology, and it has strong ecclesiological implications..." - The first part of this sentence is wrong, emphasizing only one dimension of Augustine's "soteriology". The second part is vague and obvious - the following qualifying clause does not provide clarity.
line 327 - "raise it" back up to God is confusing. Generally, as this author notes implicitly later, this would be 'transformation' not 'raise back'.
line 334 - is it a "great reversal"?
line 348-349 - "Only a book of prayers and songs..." This is an overstatement - and not true for Augustine.
line 366-367 - The first sentence of this paragraph is not clear. What is meant? Part of Augustine's "larger" theology? And why is "corpus of writings" here?
line 371 - is it 'becomes' and 'become' or 'is' and 'are'? The treatment throughout of the Totus Christus is very superficial.
The last sentence before section 5 - lines 387-388 - leads to an abrupt assertion about preaching. This needs to be smoothed out. An ideal reference for this is found in the proem to the Confessions.
line 395 - "grows up into Christ" - the author needs to clarify. This is only accurate in one sense of the church, not in proper theology of the Totus Christus.
Comments on the Quality of English LanguageStyle is halting at times.
Some minor issues: line 30 - Elysian is a poor modifier here.
line 36 - 'from which' is odd here for two reasons - the use of 'that' as the previous relative pronoun - and in relation to the clause. Perhaps re-word final clause.
line 151-152 - "something this preacher once told" - this is very confusing. I cannot tell who the 'this' is.
line 167 - "niche"?
line 189 - Why "Burns" without first name?
line 433 - 'estrangement' is, it seems, the wrong word.
Author Response
I want to thank you for your careful reading of my work and for making many careful critiques and suggestions. It is clear that you have expertise in the field of Augustine studies, and I am grateful that this work has passed over your desk in order that it might be made stronger. In all sincerity, thank you.
I have attempted to incorporate your suggestions into the paper. I have reread and edited the paper to a degree to which I think is appropriate in light of your suggestions, as well as the other reviewers'. There are several suggestions that you made which, as you can see in the revised manuscript, I have taken and made necessary adjustments.
You remark that the essay is "standard and very general" and the engagement with scholarship is limited and narrow. I think those are fair statements, and I own them. The primary intent of the work is to impact preachers and homileticians, to introduce these readers to aspects of Augustine and the history of preacher which they have likely not considered before. Because of that intent and the assumed readership, the essay does not attempt to do everything and sometimes remains general or standard rather than diving more deeply into any particular facet of Augustine's preaching. The value of the essay (as I see it) is not only in introducing these concepts to those who might not be familiar with them, but in integrating them with preaching today. Thus, the final section about preaching and formation is the most significant and the "so what" of the work. In order to make that more clear, I have elongated that section and added some other sources.
Regarding some of the suggestions for content insertions, I have taken some and left others aside, simply because my space in this article is limited and certain material (in my estimation) would not have contributed in a significant way to my primary thesis and purpose.
As I edited my essay, I tried to use language that did not overstate of cast things too simply. I also thought that statement was fair.
Again, thank you for your careful reading of my work, and your honest feedback concerning it. I hope that the changes I have made to the piece and left highlighted show that I have attempted to take these critiques and heed them to improve the article.
Reviewer 3 Report
Comments and Suggestions for Authors
The Author has chosen a very interesting and at the same time challenging topic, which has been dealt with in an exceptionally well-argued and compelling manner. Not only is the text interesting from a scholarly point of view, but because it is simply very well written, it engages and draws the reader in. This is a rare quality in scholarly texts and one that should definitely be appreciated. At first, I had doubts about the Author's use of the term 'grammar' in the context presented in the article, but these were not confirmed in the further reading the text. The author explains and defends the use of the term with considerable precision and in a convincing manner. The reader further discovers a broader and more profound meaning of this "grammar" while delving deeper into the text. The reason behind all this is that the text itself has a well-thought-out structure that clearly and cogently supports the underlying reasoning.
I would like to make a suggestion regarding the structure of the text: perhaps it would be worth dividing the fourth paragraph into smaller sections, which would at the same time highlight the main components (hence rules or principles) of the Augustinian "grammar" of the herumenutics of Christian life emerging from the Psalms.
Although there is no explicitly expressed conclusion in the text, it can be assumed that the last (i.e. fifth) paragraph summarises and transposes the historical analysis into the reality of contemporary homiletics and preaching. This should be considered a sufficient and convincing form of conclusion.
In conclusion, it should be stated that the article presented for review makes an important contribution to the development of scholarly research on the issue addressed and should therefore be published in its present form.
Author Response
I want to thank you for your reading of my work, and the tremendous encouragement you have given me by your appraisal. As I read your comments, it seemed to me that you "heard" and appreciated the article in the ways I that intended. So, I thank you. You will be able to see in the revised version that I took some suggestions of reviewers in order to improve the piece. I have left the fourth section as it was, and with the elongation of the fifth section, I do not think there is as much imbalance as perhaps there was before in the length or foci of and between the sections. I hope you will find this appropriate. Again, thank you for your reading this work, and thank you for your feedback.
Round 2
Reviewer 1 Report
Comments and Suggestions for Authors
The additions, especially at the beginning and at the end of this article, clarify the purpose and aim of article. Thus, the article is a fine example of the application of patristic texts to contemporary homiletics.
I agree to the clarification of the idea of deification. Augustine clearly thought that redemption is participation in Christ. According to Augustine, it is important to distinguish between God and humans, and between Christ and Christians. Thus, one can use the term "deification" as a synonym for "participation in Christ", however, Augustine did not think (as did some Greek theologians) that Christians would become like God.
Reviewer 2 Report
Comments and Suggestions for Authors
For additional changes, use your judgment as to what seems suitable.